# Genome-Wide Identification of Proline Transporter Gene Family in Non-Heading Chinese Cabbage and Functional Analysis of *BchProT1* under Heat Stress

**DOI:** 10.3390/ijms25010099

**Published:** 2023-12-20

**Authors:** Jun Tian, Kaizhen Chang, Yingxiao Lei, Shuhao Li, Jinwei Wang, Chenxin Huang, Fenglin Zhong

**Affiliations:** College of Horticulture, Fujian Agriculture and Forestry University, Fuzhou 350002, China; 1200306008@fafu.edu.cn (J.T.); ckz294859036@163.com (K.C.); 18006916238@163.com (Y.L.); being@fafu.edu.cn (S.L.); 3175206031@fafu.edu.cn (J.W.); 3210330023@fafu.edu.cn (C.H.)

**Keywords:** ProT, non-heading Chinese cabbage, genome-wide analysis, heat stress

## Abstract

Non-heading Chinese cabbage prefers cool temperatures, and heat stress has become a major factor for reduced yield. The proline transporter protein (ProT) is highly selective for proline transport, contributing to the heat tolerance of non-heading Chinese cabbage. However, there has been no systematic study on the identification and potential functions of the *ProT* gene family in response to heat stress in non-heading Chinese cabbage. We identified six *BchProT* genes containing 11–12 transmembrane helices characteristic of membrane proteins through whole-genome sequencing. These genes diverged into three evolutionary branches and exhibited similarity in motifs and intron/exon numbers. Segmental duplication is the primary driving force for the amplification of *BchProT*. Notably, many stress-related elements have been identified in the promoters of *BchProT* using cis-acting element analysis. The expression level of *BchProT6* was the highest in petioles, and the expression level of *BchProT1* was the highest under high-temperature stress. Subcellular localization indicated their function at cell membranes. Heterologous expression of *BchProT1* in Arabidopsis plants increased proline transport synthesis under heat-stress conditions. This study provides valuable information for exploring the molecular mechanisms underlying heat tolerance mediated by members of the *BchProT* family.

## 1. Introduction

Proline plays a pivotal role as an osmoregulatory substance in plants [1]. With its high water potential and low molecular weight, proline serves as a stabilizing agent for polypeptide chains [2]. This amino acid effectively regulates cell osmotic pressure, sustains the pressure potential of macromolecular substances, and preserves protein integrity [3]. The accumulation of proline in different plant organs varies with abiotic stress conditions [4], being attributed not only to enhanced synthesis and reduced degradation but also to the active uptake of proline, which suggests the presence of specific amino acid transporters [5]. In mature pollen tubes and seeds experiencing water stress, chloroplast tissue proline remains largely unaffected by both synthetic and degradative processes; instead, levels primarily rely on exogenous proline transport.

The plant proline transporter proteins family (ProTs) was the first family of proteins found to transport proline, and it belongs to the AAPs (amino acid/growth factor permease family) of the AATs (transporter protein superfamily), with the protein structural domains of amino acid transporter proteins [6]. They play an indispensable role in various processes of plant growth and development, such as long-distance amino acid transport, response to pathogens and abiotic stresses, and crop quality improvement [7,8]. Plant *ProT* was first isolated from *Arabidopsis thaliana*, where three family were identified, *AtProT1*, *AtProT2*, and *AtProT3*. These are expressed in several tissues of *Arabidopsis thaliana*, but there is a clear tissue specificity in expression [9]. As the study on ProT continued, it was found that stress-induced proline accumulation was also affected by uptake and translocation. ProT has different substrate specificities in different plants and can regulate the concentration of soluble solutes in cells by altering the rate of uptake or the release of compounds such as proline to improve plant stress tolerance [10,11,12]. For example, proline accumulation in the elongation zone of maize roots under low water potential conditions is mainly from proline translocation rather than synthesis [13]. *HvProT* is abundantly expressed in the root tip, especially in the root crown and cortical cells, under salt stress [14]. In soybean studies, the expression levels of *GmProT1* and *GmProT2* in leaves were strongly induced under stress conditions. Also, the overexpression of these two genes resulted in Arabidopsis plants being more tolerant to abiotic stress than wild-type plants, and they played a positive role in Arabidopsis by affecting the proline synthesis and response system [15]. This suggests that *ProT* is important for maintaining proline homeostasis under stress conditions.

Non-heading Chinese cabbage (*Brassica campestris* ssp. *chinensis*), a biennial cruciferous herb [16], is believed to have originated in China. Leaves are the main photosynthetic organ, and it is classified as a diploid plant with a chromosome number of 2n = 2x = 20. This particular subspecies of *Brassica* belongs to the Cruciferae family and possesses a high dietary fiber content, rendering it highly valuable in terms of nutritional benefits. However, the cultivation of non-heading cabbage faces challenges in cold and cool climates due to elevated ambient temperatures, which significantly impede the annual yield. To cope with high-temperature stress, Chinese cabbage accumulates metabolic substances such as proline, betaine, mannitol, etc. These primary metabolites are directly involved in osmoregulation to maintain cellular morphology, protect enzyme activity and protein structure, and stabilize metabolic functions [17]. Among these osmoregulatory substances, proline is the most widely distributed and is usually characterized by high solubility, a lack of toxicity, and accumulation in cells, without interfering with normal intracellular biochemical reactions [18].

To gain a comprehensive understanding of the pivotal role played by *ProT* in the proline transport pathway and to characterize the *ProT* gene family under heat stress, this study conducted a genome-wide identification of physicochemical properties, evolutionary relationships, structural and functional characteristics, and promoter element enrichment analysis for members of the *ProT* gene family in the non-heading Chinese cabbage genome. Expression patterns of these genes were examined across different tissues and under heat-stress conditions, and a functional characterization of major stress response genes was performed. This study establishes a foundation for comprehending the structural properties and functions of *BchProTs*, provides genes for further investigation into heat resistance mechanisms in non-heading Chinese cabbage, and offers a theoretical basis for enhancing cultivation practices, improving quality, and breeding superior varieties.

## 2. Results

### 2.1. Characterization and Analysis of BchProTs in Non-Heading Cabbage

A total of six *ProT* genes were identified in the non-heading cabbage genome by performing BLASTp searches with the three *ProT* amino acid sequences of *Arabidopsis thaliana* and conducting Pfam analysis. The genes were renamed *BchProT1-6*. Physicochemical characterization of the *BchProT* gene family (Table 1) revealed that the coding sequence lengths (CDS) of the six *BchProT* genes ranged from 1281 bp (*BchProT2*) to 1794 bp (*BchProT3*), with amino acid lengths ranging from 427 (*BchProT2*) to 598 (*BchProT3*). The molecular weights (MWs) ranged between 46.51 kDa and 66.20 kDa and the isoelectric points (pIs) from 8.62 to 9.38, except for BchProT6, which has twelve transmembrane helices (TMHs). All other members contain eleven TMHs, and subcellular localization predictions indicated that all non-heading cabbage *ProTs* are located at the cytoplasmic membrane, suggesting that *BchProTs* have a functional basis for amino acid transport.

### 2.2. Chromosome Localization and Analysis of Gene Duplication in the BchProT Gene Family

The six *ProT* genes of non-heading cabbage are unevenly distributed across the four chromosomes: B03 and A09 each carry one gene; two genes are located on chromosome A04 and two on chromosome A05. Intraspecific covariance analysis was conducted to further investigate the phylogenetic relationships among members of the *BchProT* gene family (Figure 1). For *BchProT*, one tandem duplication gene pair (*BchProT4/BchProT5*) and five segmental duplication gene pairs (*BchProT1/BchProT3*, *BchProT1/BchProT5*, *BchProT2/BchProT3*, *BchProT2/BchProT5*, *BchProT3/BchProT6*) were detected, indicating segmental duplication as the primary mode of amplification of *BchProT* genes. Additionally, it was observed that chromosomes with low gene density had a higher *BchProTs* ratio; thus, they were used to analyze evolutionary selection pressure on colinear gene pairs in the genome of non-heading cabbage. The Ka/Ks (Appendix A) values for all six colinear genes were less than one, suggesting purifying selection as the main driving force in the evolutionary process for these genes.

Interspecies covariance analysis (Figure 2) among Arabidopsis, bok choy, and non-heading cabbage was conducted for *ProT* genes. Ten covariant gene pairs were formed by the five *ProT* genes of non-heading cabbage and the two *ProT* genes of Arabidopsis, mainly on chromosome 4 of non-heading cabbage. Additionally, twenty-three covariant gene pairs were observed among the five *ProT* genes of cabbage. This suggests a high degree of homology between non-heading cabbage and both Arabidopsis and cabbage, indicating potential similar biological functions for these genes. These findings provide directions for exploring *BchProT* genes in terms of species evolution and function.

### 2.3. Conserved Motifs and Gene Structure of the BchProT Gene Family

A phylogenetic tree of the ProT proteins from non-heading cabbage was constructed, and the conserved motifs and gene structures of the *BchProT* proteins were further analyzed (Figure 3). Thirteen different motifs were identified for the *BchProT* gene family, showing a high degree of similarity in both the type and number of protein motifs among members. However, certain proteins exhibited functional differences due to the presence or absence of specific motifs; for example, motif 12 is not present in *BchProT1*, and motif 13 is only found in *BchProT3* and *BchProT4*, which are located within the same branch. The members of the *BchProT* family also differ in terms of their untranslated regions (UTRs) and coding sequences (CDSs), with varying numbers ranging from 6 to 10 CDSs.

### 2.4. Phylogenetic Analysis of ProT Genes in Multiple Species

To assess the evolutionary relationship between non-heading cabbage and members of the *ProT* gene family in the model plants Arabidopsis thaliana, tomato, rice, and wheat, a phylogenetic tree was constructed using the NJ method based on the full-length amino acid sequences of 28 *ProT* genes (Figure 4). The *BchProT* gene family can be classified into three major evolutionary branches, similar to *Arabidopsis thaliana*: *AtProT1*, *AtProT3*, and *AtProT2*. *BchProT1* and *BchProT2* are in the same branch as *AtProT2*; *BchProT3* and *BchProT4* are in the same branch as *AtProT1*; and *BchProT5* and *BchProT6* are in the same branch as *AtProT3*. Tomato *ProT* forms a separate evolutionary cluster, suggesting that *SlProT* is specific to cellular metabolism within species. As a dicotyledonous plant, non-heading cabbage is more abundant than monocotyledonous plants, indicating strong natural selection acting on *ProT* genes during evolution.

### 2.5. Predictive Analysis of Cis-Acting Elements within the Promoter Region of the BchProT Gene Family

Cell signalling is an important communication process that regulates basic activities and interactions with other cells and the environment [19]. Transcriptional regulation, as mediated by transcription factors interacting with specific DNA elements known as cis-acting elements, is a major component of the plant cell signalling system. *ProT*, a type of transporter protein crucial for plants, is also regulated by the binding of transcription factors to its promoter region in response to changes in the external environment and metabolic processes related to growth and development. A total of 826 cis-acting elements were identified by extracting the 2000 bp sequence upstream of *BchProT* coding sequences (CDSs) (Figure 5). These elements include core promoter elements associated with promoter transcription (70.58%), such as the CAAT-box and TATA-box, as well as stress response-related elements (15.25%) (Appendix A), such as ARE for anaerobic induction, LTR for low-temperature response, MBS for drought induction, STRE for heat stress response, TC-rich repeats involved in defense and stress response, and common stress response element MYB/MYC binding sites along with a W-box/WRE3/WUN-motif associated with biotic stress. The remaining cis-elements can mainly be classified into those associated with light responses (6.78%), hormone responses (4%), and growth/development regulation (3.39%). The different types of cis-elements indicate the functional diversity of *BchProT* genes, which play significant roles, especially in abiotic stress responses.

### 2.6. Analysis of BchProT Gene Family Expression in Non-Heading Cabbage under Heat Stress

Owing to the presence of a large number of stress-responsive cis-acting elements in the promoter region of *BchProTs*, we performed qRT-PCR analysis to investigate the tissue-specific expression patterns of *BchProTs* in leaves (Figure 6), petioles, and roots (A), as well as their response to temperature stress (B), to gain further insights into the stress response mechanism in non-heading cabbage. Our results revealed that all *BchProTs*, except for *BchProT2*, were expressed in the leaves, petioles, and roots of non-heading cabbage. Notably, *BchProT1* and *BchProT3* exhibited higher expression levels in leaves than in petioles; conversely, *BchProT4* and *BchProT5* showed higher expression levels in petioles than in leaves and roots. Moreover, we found elevated expression levels of *BchProT6* in both petioles and roots, underlining its potential role in long-distance proline transport. Leaf expression levels of *BchProT1* and *BchProT4* showed significant increases of 1.6-fold and 1.4-fold, respectively, under high-temperature stress. Conversely, the expression level of *BchProT3* remained unchanged under high-temperature stress, whereas that of *BchProT5* and *BchProT6* was significantly diminished.

### 2.7. Conserved Motifs and Gene Structure of the BchProT Gene Family

The recombinant plasmid pCAMBIA1302-35S-BchProT1-GFP was transformed into *Agrobacterium rhizogenes* GV3101, and tobacco leaves were transformed, with pCAMBIA1302 null as the control. After 2 days, fluorescent proteins were observed using confocal microscopy to investigate *BchProT* response to high-temperature stress in non-heading cabbage. *BchProT1*, which exhibited significant upregulation under high-temperature stress, was selected for further study. Laser confocal microscopy revealed pCAM1302 green fluorescence throughout the intact cells. Conversely, the fusion protein produced by pCAMBIA1302-35S-BchProT generated green fluorescence at the cytoplasmic membrane of the tobacco cells (Figure 7). This indicates that *BchProT1* localizes to the cytoplasmic membrane of cells, which supports subcellular localization predictions. Based on these predictions and observations, it can be speculated that *BchProT1* exists in the plasma membranes as part of a long-distance transport system and mediates amino acid translocation across cell membranes.

### 2.8. Overexpression of BchProT1 in Arabidopsis and Its Response to Elevated Temperature

To further explore the role of *BchProT1* in high-temperature stress, we constructed an overexpression plasmid and obtained *BchProT1*-overexpressing transgenic Arabidopsis via the floral dip method (Figure 8). Under high-temperature stress, WT Arabidopsis lines showed leaf curling and wilting compared with transgenic Arabidopsis. Proline and MDA are considered to be reliable indicators of environmental stress severity in plants, and the MDA content of transgenic Arabidopsis was significantly lower than that of WT Arabidopsis. The proline content was higher than that of WT Arabidopsis under high-temperature stress. These results indicate that the overexpression of *BchProT1* in Arabidopsis resulted in more tolerance to high-temperature stress than in WT plants.

## 3. Discussion

Proline accumulation is a metabolic adaptation mechanism in many plants under different environmental stresses, and the regulation of proline metabolism has been summarized by scholars at home and abroad. However, in in-depth studies, it was found that stress-induced proline accumulation is also affected by translocation [20]. *ProTs*, the protein family initially identified for proline transport, play a pivotal role in facilitating proline transport to various organs while also transporting GABA and glycine betaine. These transporters significantly contribute to crop production. Involvement of the *ProT* gene family in proline metabolism has been extensively studied in numerous plants, such as *Arabidopsis thaliana* [21], soybean [15], and tomato [22]. However, the function of the *ProT* gene family in non-heading Chinese cabbage has not yet been systematically identified and studied. With the continuous improvement in the genomic information of non-heading Chinese cabbage, we have analyzed the identification, phylogeny, and gene structure of the *ProT* gene family of non-heading Chinese cabbage, as well as the expression profiles and the gene function at high temperatures, which can help to provide a new way of thinking for a more comprehensive understanding of *BchProTs* genes.

The whole genome of non-heading Chinese cabbage was found to contain six members of the *ProT* gene family, which is a higher number than the three members in *Arabidopsis thaliana* and four in tomato [22]. This discovery indicates the presence of a multigene family for *ProT* in non-heading Chinese cabbage. The size of the *ProT* gene family in *Arabidopsis thaliana* ranges from 1308 to 1326 bp, encoding amino acids ranging from 436 to 442 aa. These genes exhibit similar physicochemical properties as those found in non-heading Chinese cabbage ProTs, suggesting that *BchProTs* are highly conserved and maintain structural integrity as well as function. A characteristic sequence feature of amino acid transport proteins is the presence of transmembrane helices, with each member of *BchProTs* containing 11–12 variable transmembrane helices, consistent with studies on tea tree and tomato *ProT*, respectively [22]. Based on these results, it can be inferred that non-heading Chinese cabbage *ProT* functions similarly to proline transport proteins in other plants and possesses proline transport capabilities, and thus *BchProTs* are typical members of the amino acid transport protein family.

Gene duplication serves as a paradigm for the occurrence, maintenance, and evolution of identical genes, playing a pivotal role in the functional diversification of multigene families [23]. The *BchProTs* analysis revealed a total of five collinear gene pairs, comprising one tandem replicating gene pair and four fragment replicating gene pairs. Fragment replication emerged as the predominant mode of expansion for *BchProTs*. Notably, all five collinear gene pairs had exhibited Ka/Ks values below one, indicating the occurrence of purification selection during the evolutionary process of *BchProTs*. This finding underscores the crucial role played by purification selection in maintaining the functional stability of genes. A total of 10 and 23 orthologous genes were identified between non-heading Chinese cabbage and cabbage, as well as Arabidopsis, respectively. In these findings, Chinese cabbage has undergone a whole-genome triplication event since its divergence from the Arabidopsis lineage [24]. The rapid polyploidization ability and genome stabilization of *Brassicaceae* adaptively evolve in rapidly changing environments, thereby sustaining overall genetic diversity over extended periods [25]. This provides a genetic foundation for the differentiation and functional diversity of *BchProTs*, enabling plants to effectively cope with various environmental stresses during the process of evolution.

A phylogenetic tree can depict the evolutionary order or formation of organisms, with connections representing genetic relationships between species or different entities [26]. Notably, among ProTs, monocotyledonous plants such as rice and wheat form an independent sub-branch. *BchProTs* and *AtProTs*, both belonging to dicotyledonous plants, are distributed together in three evolutionary branches, indicating their close relationship. Furthermore, a functional analysis of *Arabidopsis ProT* genes can serve as a valuable reference for investigating the function of *BchProTs* genes in non-heading Chinese cabbage. This suggests that *ProT* genes may have evolved subsequent to the differentiation of monocotyledonous and dicotyledonous plants, a pattern similar to phylogenetic clustering results observed in tomato ammonium transporter gene families in both types of plants [27]. Through the systematic phylogenetic analysis of conserved motifs and gene structures within the *BchProT* family, members located on the same evolutionary branch exhibit comparable motif distributions. Among these members, Motif12 and Motif13 demonstrate specificity across different *BchProTs* members, which is associated with conferring specific protein functions. Four tomato *ProT* genes in separate branches with two exons each exhibiting relatively conserved characteristics occur regarding intron–exon numbers [22], lengths, and positions among six *BchProTs* genes, suggesting strong natural selection during their evolution process. These differences likely contribute to distinct functionalities observed in tea trees [28] compared to tomato [22] studies.

At the promoter, cis-acting elements interact with trans-acting factors to tightly regulate gene expression. The promoter sequence of *EgProT1* in oil palm was found to contain a significant number of stress response cis-elements [29]. In this study, we detected 826 cis-acting elements within *BchProT* promoter regions. Among them, 15.25% are categorized as stress-related elements, making it the largest category apart from core promoter elements. Notably, the MYB element was identified as a major stress-responsive element, highlighting its potential role in transcription factors and the modulation of transcriptional expression during the stress response.

Expression levels in several members of the *ProT* family are often associated with stress conditions or high Pro concentrations [9]. It was previously reported that *AtProT1* in Arabidopsis exhibits high expression levels in stems, roots, and flowers, indicating its crucial role in long-distance proline transport. *AtProT2* is predominantly expressed in roots, while *AtProT3* shows tissue-specific expression patterns in leaves and stems [9,21]. The expression level of *SlProT1* was found to be the highest in seeds and pollen with elevated proline concentrations [11]. *OsProTs* in rice exhibit significant expression levels in both stems and leaves [30]. Soybean *GmProTs* were tissue-specifically expressed in leaf stems and roots [15]. Real-time fluorescence quantitative PCR analysis demonstrated temporal–spatial specificity of *BchProTs* expression. *BchProT1* and *BchProT3* exhibited higher levels of expression in leaf blades, which may be associated with maintaining stable cell morphology or facilitating the distribution of proline contents among tissues under external environmental changes. *BchProT4* and *BchProT5* displayed the highest expression levels in petioles, suggesting their potential involvement primarily in proline transport within these structures. *BchProT6* showed high expression levels in petioles and roots, which is likely associated with long-distance proline transport. This observation highlights the differential expression of *BchProTs* across various organs, potentially indicating their tissue-specific transport specificity. Plant *ProTs* have been widely implicated in the environmental stress response. In this study, *BchProT1* and *BchProT4* exhibited significantly higher responsiveness to elevated temperatures than other members. This is consistent with previous findings on *GmProT1* and *GmProT2* in soybean [15], with upregulation under stress conditions and amelioration of damage caused by salt and drought stresses. The observed pronounced upregulation of *BchProT1* expression under heat stress suggests its pivotal role in Chinese cabbage’s response to temperature stress, highlighting its significance in maintaining cellular homeostasis and facilitating proline transport and distribution among various tissues as a response to external stresses.

Proline is produced by ProTs transporters through the plasma membrane, degradation of proline occurs in mitochondria, and synthesis of proline occurs in the cytoplasm, but accumulates in the cytoplasm and chloroplasts under stress conditions. However, under stress conditions, proline accumulates in both compartments, indicating the existence of a plasma membrane transport system for proline [31]. In this study, *BchProT1* contained abundant transmembrane structural domains, and the subcellular expression of *BchProT1* was localized to the cytoplasmic membrane. Studies in Arabidopsis have demonstrated that *AtProTs* mediate the uptake of compatible solutes in extracellular spaces [9]. In barley, *HvProT* functions as a high-affinity transporter protein for proline [32], and experiments using ^15^N-labeled proline absorption have shown that OsProT1 and other functional absorptive transporters are involved in proline uptake [33]. This suggests that *BchProT* likely possesses similar transport functions in Chinese cabbage.

The presence of stress-responsive elements in the ProT promoters of non-heading Chinese cabbage, along with the high expression and subcellular localization results of *BchProT1* under heat stress, suggests its role in responding to heat stress. To investigate its function, we introduced *BchProT1* into *Arabidopsis thaliana* using the floral dip method and observed that *BchProT1* did not affect normal Arabidopsis growth. However, the ectopic expression of *HvProT* in Arabidopsis inhibited its growth [14], indicating functional differentiation among different members and crops within the *ProT* gene family. The overexpression of rice *ProT* genes significantly enhanced rice’s stress tolerance, while the overexpression of mangrove *ProT* genes improved salt tolerance in rice [34]. Additionally, overexpressing spinach beetroot (Beta vulgaris) *ProT* genes in Arabidopsis increased its salt tolerance under saline conditions [35]. A close relationship was found between ProT genes and plant stress resistance. Malondialdehyde (MDA) disrupts cell membrane structure and function, and higher accumulation levels indicate greater damage severity. In this study, the overexpression of *BchProTI* in Arabidopsis resulted in a significantly lower MDA content than that in wild-type plants under heat-stress conditions. Moreover, the free proline content was significantly higher due to the efficient transport ability induced by overexpressed *BchProTI* in leaves to accumulate proline under stress. This led to an increase in the intracellular proline content and alleviation of the harm caused by heat stress. These findings further support the association between *BchProTI* and plant thermotolerance.

## 4. Materials and Methods

### 4.1. Identification of ProTs Members of Non-Heading Chinese Cabbage

A comparative analysis was conducted based on the whole genome data of non-heading Chinese cabbage from the NCBI database (https://www.ncbi.nlm.nih.gov/, accessed on 23 December 2022) with the accession number PRJNA735552 [36]. The amino acid sequences encoded by *ProT* genes in *Arabidopsis* were obtained from the TAIR database (https://www.Arabidopsis.org/, accessed on 23 December 2022). Known protein sequences of three *ProT* genes in *Arabidopsis* were used as query sequences to search for counterparts in the non-heading Chinese cabbage genome using BLASTP by employing a default threshold of e^−10^. Hidden Markov Model (HMM) configuration files for the protein domain PF01490 were obtained from the Pfam website (http://pfam.xfam.org/, accessed on 28 December 2022), and HMMER v3.0 software was used to identify all potential *ProT* genes. The conserved protein domains were cross-checked against the NCBI Conserved Domain Database (https://www.ncbi.nlm.nih.gov, accessed on 3 January 2023). After eliminating redundantPro (http://www.ebi.ac.uk/interpro/scan.html, accessed on 6 January 2023), the presence and integrity of conserved structural domains in *ProT* genes were confirmed. After all *BchProTs* gene family members within the non-heading Chinese cabbage genome were identified, nomenclature assignment and the compilation of relevant information such as full-length cDNA, coding sequence length, gene structure, and protein product were performed.

The gene structure of membrane regions (TMHs) was analyzed using tools provided by the GSDS website (http://gsds1.cbi.pku.edu.cn/index.php, accessed on 15 January 2023), with TMHMM Server v2.0 (http://wwwbs.dtu.dk/services/TMHMM/, accessed on 15 January 2023) used for prediction. Furthermore, biochemical parameters such as the theoretical isoelectric point (pI) and molecular weight (MW) were calculated using the ExPASy database (https://web.ExPASy.org/compute_pi/, accessed on 15 January 2023). To assess similarity and identity among *BchProTs* members, global sequence alignment was performed utilizing EMBOSS’s needle tool (http://emboss.bioinformatics.nl/, accessed on 15 January 2023).

### 4.2. Analysis of BchProTs Duplication and Selection Pressure

To investigate the genetic evolution of *ProT* genes and their relationship with gene duplication in non-heading Chinese cabbage and Chinese cabbage (*Brassica rapa var.*), we obtained whole-genome data of Chinese cabbage from BRAD. Gene duplications were detected using the MCScanX program [37] based on the non-heading Chinese cabbage and heading Chinese cabbage genomes. TBtools [38] was utilized to visualize the chromosomal localization and duplicated regions of all *BchProTs*. TBtools was employed to visualize chromosome replication in non-heading Chinese cabbage. KaKs-Calculator 2.0 [39] was used to determine synonymous (Ks) and nonsynonymous (Ka) substitution rates (Ka/Ks) for each pair of *BchProT* gene duplications to evaluate the functional impact of gene duplication.

### 4.3. Multiple Sequence Alignment and Phylogenetic Analysis

To elucidate the evolutionary relationship between members of the *ProT* gene family in non-heading Chinese cabbage and other plant species, we classified *ProT* gene family members in non-heading Chinese cabbage and Chinese cabbage. Subsequently, we compared the protein sequences of ProT genes and *BchProTs* with those from rice (*Oryza sativa* L.) [40], wheat (*Triticum aestivum* L.) [41], and tomato (*Solanum lycopersicum* L.) [22] using MAFFT (https://www.ebi.ac.uk/Tools/msa/mafft/, accessed on 13 February 2023). Alignment was performed using MAFFT [42], followed by the removal of redundancy, and a tree model was constructed using EGA7 software. To enhance visualization, the phylogenetic tree was refined using the iTOL tool (http://itol.embl.de, accessed on 13 February 2023). Additionally, conserved protein motifs were identified for these genes utilizing the Multiple EM for Motif Elimination (MEME) program with motif widths ranging from 6 to 200 residues with the maximum number of motifs set at 20. DNAMAN (DNAMAN Version 9). software was employed to analyze conserved amino acid sequence motifs as well as transmembrane regions.

### 4.4. Cis-Acting Regulatory Element Analysis of Putative Promoter Sequences

The sequence of a region 2000 bp upstream of the *BchProTs* start codon (ATG) was extracted from the non-heading cabbage genome using the TBtools program, and the region was analyzed using PlantCARE (http://bioinformatics.psb.ugent.be/webtools/plantcare/html/, accessed on 17 February 2023) to identify the cis-acting elements of all *BchProTs* genes.

### 4.5. Plant Materials, Heat Stress, RNA Extraction, and qRT-PCR Analysis

Leaves, petioles, roots, and leaves subjected to high-temperature stress by treatment at 40 °C for 12 h were collected from normal-growing ‘Jinpin yixia’ non-heading cabbage, while control samples were collected at 0 h. Total RNA was extracted using a FastPure Plant Total RNA Isolation Kit (Vazyme, Nanjing, China), and cDNA first-strand syntheses were performed using a HiScript III 1st Strand cDNA Synthesis Kit (Vazyme, Nanjing, China). Real-time fluorescent quantitative PCR primers were designed using Primer Premier 5.0 software, with *BchActin* as the internal reference (primer sequences are listed in Appendix A). Real-time fluorescence quantitative PCR was conducted using a Light Cycler 96 real-time SYBR Taq protocol. Each sample involved three independent biological replicates with three technical replicates, and expression levels were evaluated by calculating (2^−ΔΔCt^).

### 4.6. Verification of BchProT1 Protein Localization

To check the prediction results of *BchProT1* protein localization, cDNA from the leaves of non-heading Chinese cabbage was used to amplify the full-length CDSs of *BchProT1* and *BchProT1* without stop codons with specific primers (primer sequences are listed in Appendix A). The target fragment was inserted in the GFP-tagged PCAM-BIA1302 vector under the control of the 35S CaMV promoter using homologous recombination, and the conventional freeze-thaw method was used to transform the plasmid into Agrobacterium tumefaciens GV3101. *Nicotiana benthamiana* plants with good growth at the age of 5 to 6 weeks were selected for transformation, followed by culture in the dark for 12 h, and then for approximately 48 h. Laser confocal microscopy was used to detect the GFP signal.

### 4.7. Generation of Transgenic Arabidopsis

*Arabidopsis thaliana* was transformed with *Agrobacterium tumefaciens* using the floral dip method. The transformed plants were placed on a tray and covered with a black plastic bag for 24 h of dark incubation. After removing the bag, normal growth conditions (25/18 °C, 75% relative humidity, 12/12 days/nights) were applied. Seeds were harvested as the T0 generation after natural ripening. Vernalized seeds were evenly sown on 1/2 MS (2.23 g/L of Phytotech Murashige & Skoog (MS) Modified Basal Salt Mixture, 10 g/L of sucrose, 10 g/L of plant agar, PH 5.8) medium supplemented with HYG antibiotics to screen for positive transformants. These seedlings were transplanted into nutrient soil and recorded as the T1 generation. Positive plants underwent individual PCR analysis, and three progeny were obtained.

### 4.8. Heat Stress Treatment

T3 and wild-type *Arabidopsis thaliana* (WT) plants were co-cultivated in growth medium at the two-leaf stage. After 35 days, samples were collected following exposure to a high-temperature treatment at 40 °C for 24 h. The proline content was quantified using the indophenol-sulfosalicylic acid method [1], and malondialdehyde (MDA) levels were assessed using corresponding kits from Solarbio (Beijing, China). Three independent biological replicates were performed.

### 4.9. Data Analysis

Data were processed and plotted using SPSS (IBM^®^SPSS^®^ Statistics version 24) and GraphPad Prism 8 (San Diego, CA, USA).

## 5. Conclusions

In summary, we identified six *BchProT* genes in the non-heading Chinese cabbage genome, and analyzed the *BchProT* family systematically and comprehensively, including gene identification, phylogenetic analysis, replication, structure, and cis-elements. It was found that the *BchProT* genes had different expression patterns in the tissues of non-bearing cabbage, and that high temperature could induce the expression of some of these genes. Among them, high temperature induced significant up-regulation of the expression of genes *BchProT1* and *BchProT4*. It was found that BchProTs genes have different expression patterns in tissues, and that high temperature induces the expression of BchProT1 and BchProT4 genes. The ectopic overexpression of BchProT1 in transgenic Arabidopsis suggests a role in its contribution to high-temperature stress tolerance. This study lays the foundation for the further exploration of the molecular mechanism of the ProT gene family in non-bearing cabbage and its role in the high-temperature stress response, and the results of this study can be used to explore the application of ProT genes in the breeding of high-temperature-tolerant non-bearing cabbage. 

## Figures and Tables

**Figure 1 ijms-25-00099-f001:**
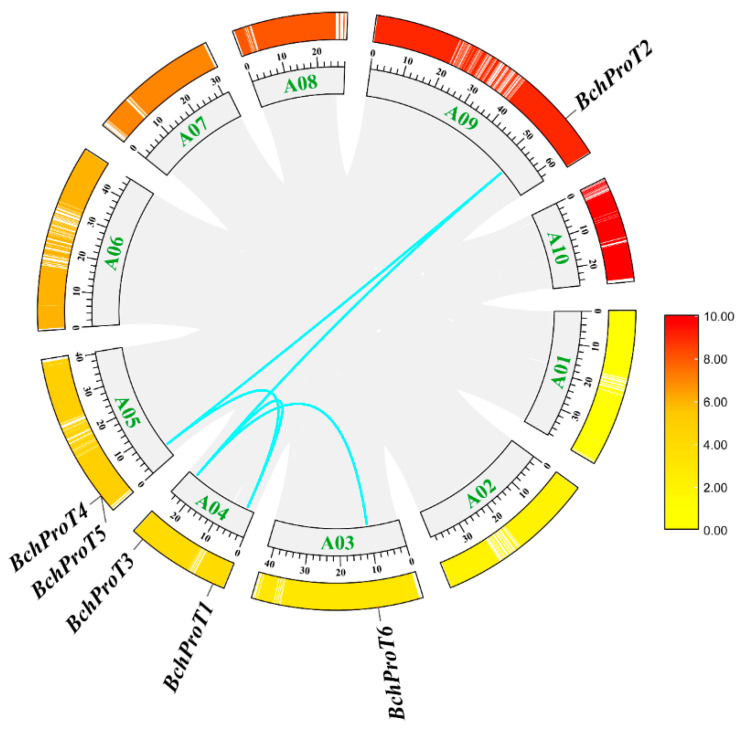
The gene–replication relationship between *BchProTs*. Outer ring representation gene density: red: higher expression; yellow: lower expression. The green fonts are 10 chromosomes A01–10, the blue lines represent collinear gene pairs, and the gray areas are collinear regions.

**Figure 2 ijms-25-00099-f002:**
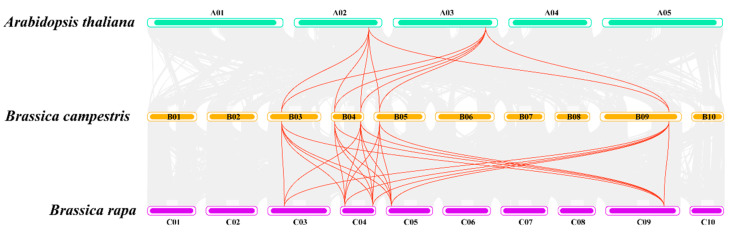
*BchProTs* collinearity analysis with Arabidopsis and Chinese cabbage. The gray lines in the background represent collinear blocks of the genome, and the red lines represent homologous ProT gene pairs.

**Figure 3 ijms-25-00099-f003:**
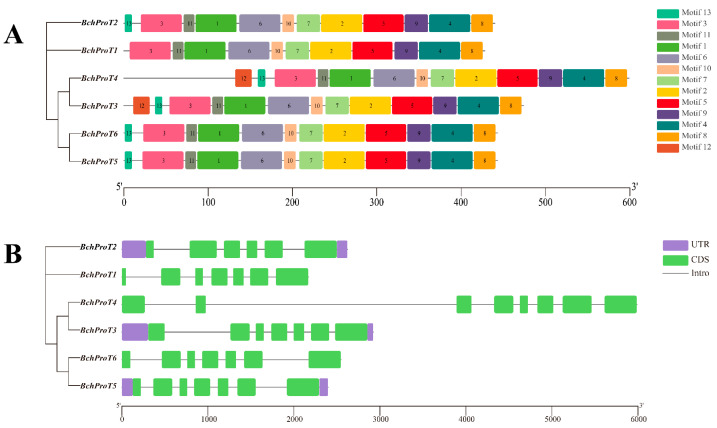
A schematic diagram of gene structure and conserved motif of the *BchProT* gene family. (**A**) Motif distribution. Twelve conserved motifs in BchProT proteins are indicated by multiple colored boxes. Different colors boxes represent different conservative motifs. (**B**) The gene structure. Purple box represents UTR, green box represents CDS, and grey line represents intron.

**Figure 4 ijms-25-00099-f004:**
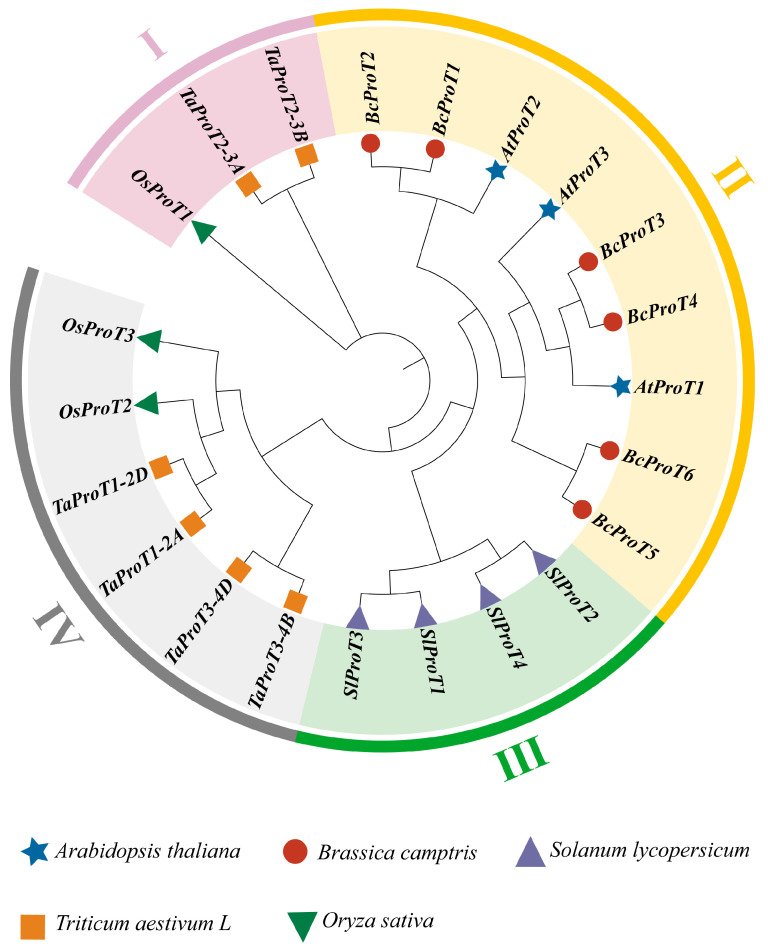
*BchProTs* with a multispecies phylogenetic tree. The outer ring of different colors represents different groups of *BchProTs* gene family. Blue star represents *Arabidopsis thaliana*, red circle represents *Brassica camptris*, purple triangle represents *Solanum lycopersicum*, yellow square represents *Triticum aestivum* L., and green triangle represents *Oryza sativa*. The number on the branch of the evolutionary tree represents the Bootstrap value.

**Figure 5 ijms-25-00099-f005:**
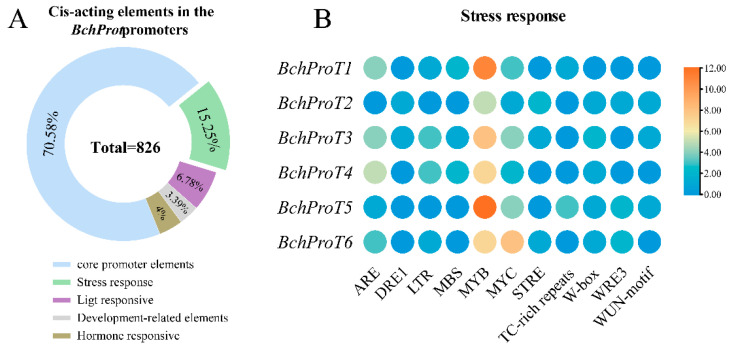
The analysis results of *BchProT* genes promoter sequence. (**A**) Functional classification of predicted cis-elements in *BchProT* promoter regions. % represents the percentage of the total number of components. (**B**) Quantity heatmap of stress responsiveness.

**Figure 6 ijms-25-00099-f006:**
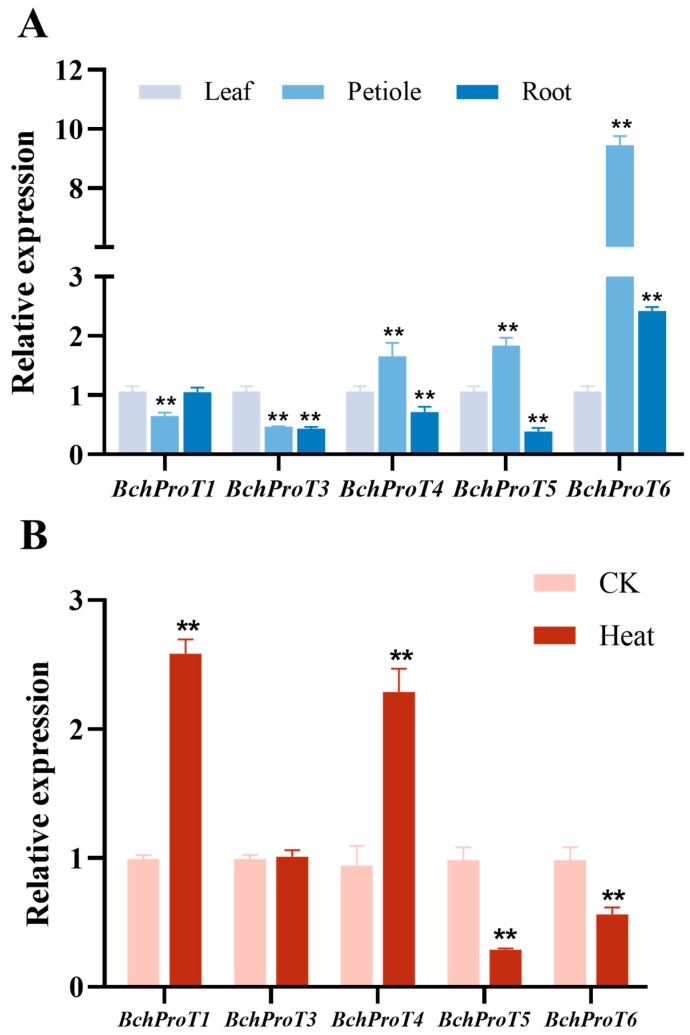
*BchProT* gene family expression. (**A**) Different tissue sites. (**B**) Heat stress (40 °C for 12 h). ** indicates extremely significant difference (*p* value < 0.01). All bars represent means ± SD (*n* = 3).

**Figure 7 ijms-25-00099-f007:**
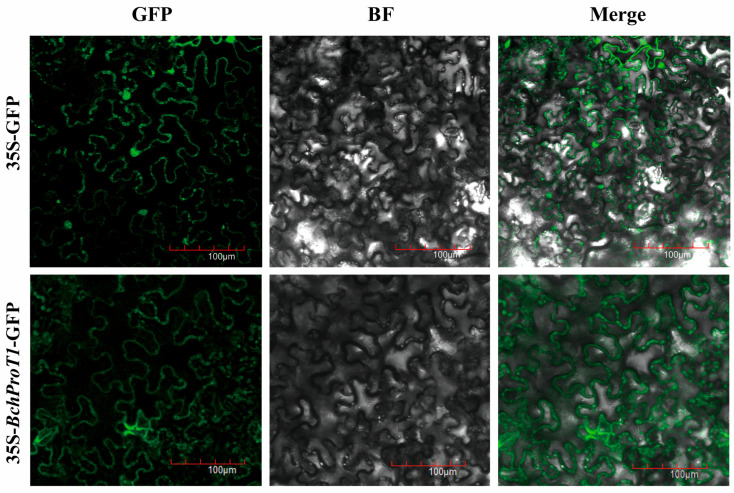
*BchProT1* subcellular localization. Note: green fluorescence (GFP); visible light (BF); merge field (Merge); and scale bar = 100 μm.

**Figure 8 ijms-25-00099-f008:**
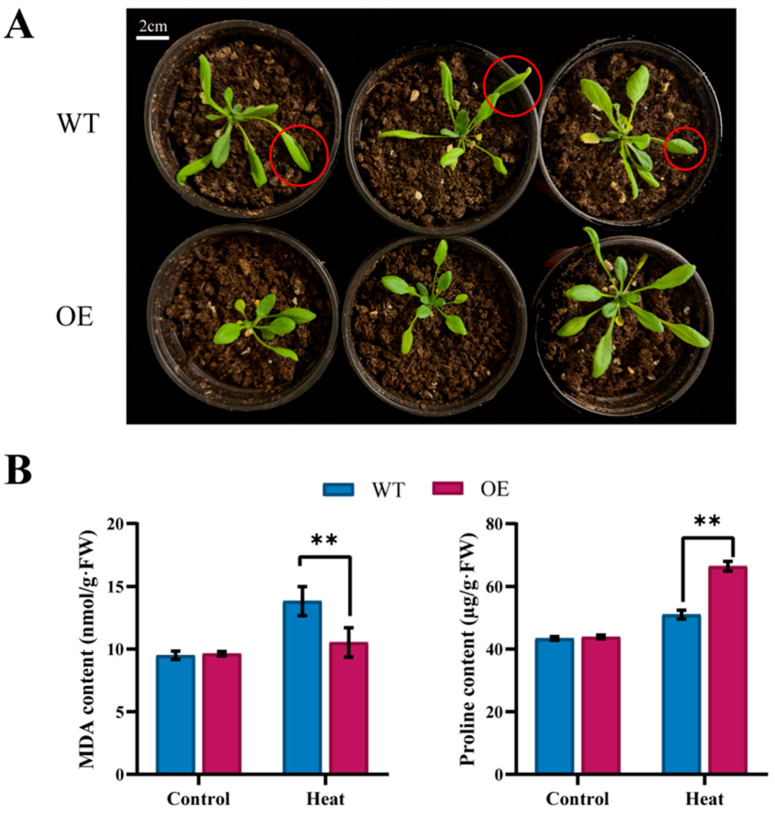
Ectopic expression of *BchProT1* in Arabidopsis. (**A**) Heat stress Arabidopsis phenotype. Red circles highlight the phenotypic changes of wild Arabidopsis under heat stress. (**B**) Heat stress Arabidopsis MDA and Proline content. WT represents wild-type plants; OE represents highly expressed transgenic lines; Heat represents (40 °C for 12 h). ** indicates extremely significant difference (*p* value < 0.01). All bars represent means ± SD (*n* = 3).

**Table 1 ijms-25-00099-t001:** Information on the *BchProt* gene family.

Gene Name	Gene ID	CDS (bp)	Amino Acids (aa)	Molecular Weight (KDa)	pI	TMHs	Subcellular Location
*BchProT1*	Bch04G004820.1	1317	439	48.01	9.38	11	plasma membrane
*BchProT2*	Bch09G053490.1	1281	427	46.51	9.19	11	plasma membrane
*BchProT3*	Bch04G032470.1	1794	598	66.20	8.62	11	plasma membrane
*BchProT4*	Bch05G006680.1	1419	473	51.81	9.01	11	plasma membrane
*BchProT5*	Bch05G006660.1	1326	442	48.33	9.23	11	plasma membrane
*BchProT6*	Bch03G022680.1	1326	442	48.53	9.38	12	plasma membrane

## Data Availability

Data are contained within the article and Appendix A.

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
