# Peer review of "Genome-Wide Identification of Proline Transporter Gene Family in Non-Heading Chinese Cabbage and Functional Analysis of *BchProT1* under Heat Stress"

_ijms, 2023, doi:10.3390/ijms25010099_

Round 1

Reviewer 1 Report

Comments and Suggestions for Authors

Dear Editors,

The manuscript entitled "Genome-wide identification of proline transporter gene family in non-heading Chinese cabbage and functional analysis of BchProT1 under heat stress” by Jun Tian et al. is within the aim and scope of International Journal of Molecular Sciences. The authors have investigated the pivotal role of ProT in the proline transport pathway as well as to characterize ProT under heat stress.

The study is attractive with interesting subject and results. The study seems to be valuable and the research methods and results are quite well documented.

Overall, the manuscript is well-written and I have minor comments. I recommend to accept it after supplementations and corrections (see attached file).

Author Response

Thank you for your valuable comments and suggestions. These comments are valuable and help to revise and improve our paper. Based on your good suggestions, we have made a lot of changes to our previous manuscript as follows. The questions raised by the reviewers are in black and the authors' responses are in red. You can review them in the attached document.

Reviewer 2 Report

Comments and Suggestions for Authors

The work was very  predictable as similar studies also exist for other plants. 

Some of the images are poor and legends for figures need more detail.

Not sure of how the studies were replicated and  stat analysis performed. 

Manuscript could be improved. 

Comments on the Quality of English Language

generally OK  but some sentences need rephrasing. 

Several times order of presentation of material could be improved.  

Author Response

We feel great thanks for your professional review work on our article. As you are concerned, there are several problems that need to be addressed. According to your nice suggestions, we have made extensive corrections to our previous draft, the detailed corrections are listed below. The black font is the question raised by the reviewer, and the red font is the author's answer.

You can review them in the attached document.
